# Bacterial Contamination of the Surgical Site at the Time of Elective Caesarean Section in Belgian Blue Cows—Part 2: Identified by 16Sr DNA Amplicon Sequencing

**DOI:** 10.3390/vetsci10020094

**Published:** 2023-01-26

**Authors:** Salem Djebala, Elise Coria, Florian Munaut, Linde Gille, Justine Eppe, Nassim Moula, Bernard Taminiau, Georges Daube, Philippe Bossaert

**Affiliations:** 1Clinical Department of Ruminants, University of Liège, Quartier Vallée 2, Avenue de Cureghem 7A-7D, 4000 Liege, Belgium; 2Murphy and Leslie Veterinary Centre (Private Practice), Muckerstaff Granard, N39AN52 Co Longford, Ireland; 3Department of Animal Production, University of Liege, Quartier Vallée 2, Avenue de Cureghem 6, 4000 Liege, Belgium; 4GIGA—Animal Facilities—ULiège—B 34, 4000 Liege, Belgium; 5Fundamental and Applied Research for Animal and Health (FARAH), University of Liege, 4000 Liege, Belgium; 6Food Microbiology, Department of Food Sciences, Faculty of Veterinary Medicine, University of Liege, 4000 Liege, Belgium; 7Faculty of Veterinary Medicine, University of Namur, Rue de Bruxelles 61, 5000 Namur, Belgium

**Keywords:** bacteriology, amplicon sequencing, microbiota, elective caesarean section, surgical site contamination, clean-contaminated surgery

## Abstract

**Simple Summary:**

To enhance the antibiotic treatment during the realisation of elective caesarean sections (CSs), a better knowledge of the bacterial composition of the surgical site is required. This study involved 13 cows of a previous dataset of 76 animals. Bacteriology was performed on swabs sampled from visceral and parietal peritoneum during the CS. DNA sequencing was performed on six samples chosen randomly among the culture-positive swabs and seven among the culture-negative swabs. No difference was observed between the microbiota composition of the samples positive or negative to bacterial culture. In fact, a high amount of bacterial DNA was identified in all surgical sites of elective CSs, and the most identified species was *Mycoplasma wenyonii*. Furthermore, the bacteria cultured were not the dominant species, since most of them were not identified by amplicon sequencing in the same sample in which they were cultured. It is assumed that the most identified bacterial DNA is due to the haematogenous contamination of the surgical site. To understand the role (protector vs. harmful) of these bacteria in the occurrence of post-operative complications and the effect of antibiotic treatment (protector vs. harmful) on the microbiota, further studies are required.

**Abstract:**

This study aimed to describe the bacterial composition of the surgical site during elective caesarean sections (CSs) using the 16Sr DNA amplicon sequencing performed in parallel to bacterial culture. The study involved 13 Belgian blue cows of a previous dataset of 76 animals. Bacteriology was performed on swabs sampled from visceral and parietal peritoneum during the CS. Amplicon sequencing was performed in six samples chosen randomly among the swabs positive for bacteriology and seven among the culture-negative swabs. A total of 2542 bacterial operational taxonomic units belonging to 567 genera were identified. The most often identified genus and species were *Mycoplasma* (44%) and *Mycoplasma wenyonii* (36%), respectively. Results showed no difference in microbiota composition between the culture-positive and -negative samples. However, a difference was observed between the bacteriology and amplicon sequencing results. Indeed, seven out of nine cultured strains were not identified by amplicon sequencing in the samples in which they were cultured. In contrast to bacteriology, amplicon sequencing unveiled the presence of bacterial DNA in all elective CSs. The most identified DNA is most likely derived from the haematogenous spread of bacteria to the surgical site. Furthermore, the cultured bacteria were not the dominant species in the sample from which they were cultured.

## 1. Introduction

Due to the large population of the Belgian blue cattle breed (BBCB), an elective caesarean section (CS) is the most common surgical intervention performed by rural vets in Belgium. Indeed, 95% of BBCB calves are born by elective CS as a result of the foeto-pelvic disproportion [1,2]. This operation is considered to be a clean-contaminated surgery [3,4]. This contamination could be exogenous by environmental bacteria [5], endogenous by the vaginal flora [3] and/or haematogenous, following the blood spread of germs [6,7].

Better knowledge of bacterial contamination of the surgical site during elective CS is needed to reduce the incidence of post-operative complications to improve the antibiotic efficacy [8,9] and to reduce the antibiotic consumption [10,11]. Unfortunately, so far the information concerning surgical site contaminations during elective CSs are limited and not accurate. The only published study dealing with the topic was performed in 1997 in Ghent University [3]. The bacterial culture performed in this study revealed the presence of several families of bacteria. Most of them are probably derived from endogenous contamination by the vaginal flora. These results are completely in contrast to those found in the first part of this study (*Bacterial contamination of the surgical site at the time of elective caesarean section in Belgian blue cows—Part 1: Identified by bacterial culture*). In fact, in this study, the most identified bacteria in the surgical site of elective CSs were aerobic Gram-negative species, most likely derived from environmental contamination. Furthermore, the number of identified bacteria is often underestimated, due to the limited sensibility of bacterial culture [12]. In addition to that, the culture of anaerobic bacteria is fastidious, since anaerobic species require special conditions [13]. As a result, more than 99% of germs present are not cultured [14]. These contradictions raise doubts about the accuracy of conclusions drawn in the study dealing with the surgical site contamination of CSs. Therefore, the utilisation of 16Sr DNA amplicon sequencing to identify the bacteria involved in the surgical site contamination might bring additional information to the results obtained by culture [15] and might contribute to implementing a better policy in the management of surgical site contamination [14,16].

The utilisation of amplicon sequencing has contributed to clarifying many issues in several areas of human medicine [17,18,19,20]. In bovine veterinary medicine, especially in the field of obstetrics and reproduction, microbiota sequencing has unveiled that the pregnant uterus is not sterile, and its microbiota composition varies according to the alimentation and the stage of pregnancy and can have an impact on the new-born’s health [21,22,23,24,25,26].

These findings, especially those describing foetal fluid’s microbiota sampled by uterus puncture during the elective CS in BBCB [23], paved the way for our study. Moreover, the multiple possible sources of surgical site contamination during the CS realisation [3,5,6] increase the likelihood of finding countless bacteria which are not detected by bacteriology [3,14,23].

Given the contradictions in culture studies, and to improve the management and the antibiotic treatment of BBCB cows during the realisation of elective CSs, more knowledge is required about the bacteria contaminating the surgical site during this operation. The present research was carried out in this context, and it aimed to describe the bacterial flora present in the surgical site of BBCB cows during the realisation of an elective CS, using amplicon sequencing. The second goal of this study was to compare the composition of the microbiota between samples showing positive and those showing negative (culture) bacteriology. Moreover, this research aimed to assess whether the cultured bacteria are the most often identified species by amplicon sequencing in positive samples.

## 2. Materials and Methods

All procedures received the approval of the Ethical Committee of Liège University (file number 2142). The cows’ owners were informed about the study and gave their consent.

### 2.1. Study Design

The data of this study were collected between February and June 2020. Two swab samples were taken in the peritoneal cavity during the realisation of elective CSs in 76 healthy BBCB cows that did not receive any treatment for at least seven months. The involved cows came from 25 distinct farms in the province of Luxembourg (Wallonia, Belgium). A first sample was kept at 4 °C and dispatched within the day to the laboratory of ARSIA to achieve bacteriology. Results of bacterial cultures were displayed in detail in part 1 of this study (*Bacterial contamination of the surgical site at the time of elective caesarean section in Belgian blue cows—Part 1: Identified by bacterial culture*). The second sample was immediately frozen and kept at −80° until all bacteriological results were fulfilled. Among the frozen samples, 13 were selected randomly to achieve amplicon sequencing in the laboratory of Food Microbiology of Liege University. Six were chosen amid samples showing a positive bacterial culture, and seven were selected among those displaying a negative bacteriology. The random choice of samples was made by the random function (Alea) of Excel (2016) software. The details of this section are displayed in Figure 1.

### 2.2. Caesarean Section Realisation

The moment of CS realisation was decided upon the vaginal palpation performed properly by the farmer and confirmed by the vet. Elective CS was carried out in each cow between the moment of the passive phase of cervical dilatation, in which the cervix is sufficiently opened to admit two to four fingers, and the phase of full cervical dilatation with intact foetal membranes [27]. The CSs were performed following the recommendations of Kolkman et al. (2007) [28] and Kolkman et al. (2010) [29]. However, small practical modifications were performed compared to these protocols, since our CSs were carried out in the field. All CSs were performed by a 7-year-experienced vet. The surgery is always carried out in upright cow imbedded in its usual place among the other cows in the late stage of pregnancy. Details of the CS realisation are displayed in part 1 of this study.

### 2.3. Sample Collection

During the CS realisation each cow was sampled twice. The samples were collected just after replacing the sutured uterus in the abdominal cavity before the closure of the abdominal wall. At this moment, swabs (STERILER^®^, Piove di Sacco, Italy) were taken by swiping a long line of 10 cm, 2 cm in parallel to the uterus suture (visceral peritoneum of the uterus) and a 10 cm long line perpendicular to and below the abdominal wall incision (parietal peritoneum).

### 2.4. Bacterial Culture

One of the sampled swabs was used for aerobic and anaerobic bacterial culture. The samples for aerobic culture were grown on Columbia agar, Gassner and Columbia/Nalidixic acid agar media (Thermo Fisher Scientific, Brussels, Belgium) at 37 ± 2 °C. Samples for anaerobic culture were grown under anaerobic conditions on Schaedler medium (Thermo Fisher Scientific, Brussels, Belgium) at 37 ± 2 °C. Two readings of each medium were performed at 18 to 24 h and 36 to 48 h of incubation. Bacterial identification was performed by the Maldi Biotyper^®^ (Bruker Daltonics, Bremen, Germany). The culture was considered “negative” if no bacterial growth was observed, “positive” when one to four bacteria species were found and “positive contaminated” when more than four species were cultured [7].

#### 2.4.1. Samples Randomly Selected for 16S Amplicon Sequencing

Following the results of the bacterial cultures performed in the first part of this study, 13 samples (6 positive cultures and 7 negative cultures) coming from 10 farms were chosen randomly among the 25 farms and 76 swabs subjected to the bacteriology. The details of the selected samples are displayed in Figure 1.

#### 2.4.2. Extraction of Bacterial DNA

Total bacterial DNA was extracted from swabs with DNeasy Blood and tissues kit (Qiagen, Belgium) using manufacturer’s protocol, modified with a bead-beating step during cell lysis [30]. Spectrophotometry (NanoDrop ND-1000, Isogen, De Meern, The Netherlands) was used for total DNA concentration evaluation.

#### 2.4.3. PCR Amplification and Product Quantification

PCR amplification targeting the V1-V3 hypervariable region of the 16S rDNA and library preparation were performed with the following primers (already linked to Illumina-adapters), forward 5′-GAGAGTTTGATYMTGGGCTCAG-3′ and reverse 5′-ACCGCGGCTGCTGGCAC-3′. PCR products were purified with the AgencourtAM Pure XP beads kit (Beckman Coulter; Pasadena, CA, USA) and subjected to a second round of PCR for indexing, using Nextera XT index primers 1 and 2. After purification, PCR products were quantified using Quant-IT PicoGreen (ThermoFisher Scientific; Waltham, MA, USA) and diluted to 10 ng/μL. Final qPCR quantification of each library sample was performed using the KAPA SYBR^®^ FAST qPCR kit (KapaBiosystems; Wilmington, MA, USA) prior to standardisation, pooling and sequencing on an MiSeq sequencer using V3 reagents (Illumina; San Diego, CA, USA). A positive control using DNA from 20 defined bacterial species and a negative control (from the PCR step) were included in the sequencing.

### 2.5. Bioinformatics Analyses

Raw sequences were processed using MOTHUR v1.41 for alignment and clustering, and the VSEARCH algorithm for chimera detection [31,32,33]. As a guideline, standard MOTHUR MiSeq SOP was used to perform the reads processing and OTU generation [34]. A clustering distance of 0.03 was used for OTU generation. The 16S rDNA reference alignment and taxonomic assignment were based on the SILVA (v1.38) database of full-length 16S rDNA sequences. From 2,616,654 raw sequences, we retained 2,525,946 sequences after cleaning (length and sequence quality) and 2,312,869 sequences with a median length of 493 nucleotides after searching and removing chimeric sequences. A rarefied table with 10,000 sequences per sample was used for taxonomic assignment and OTU clustering. Good’s coverage estimator was used as a measure of sampling effort for each sample, with an average value of 99.77%.

Negative controls, as a measure of determining erroneous results due to contamination, were not sequenced, as there was no detectable amplification product in the samples.

### 2.6. Data Analysis

A statistical analysis of the overall scores using a paired *t*-test was used. High-throughput sequencing (NGS) was performed and used to assess alpha diversity (richness estimation—Chao1 estimator; microbial biodiversity—Simpson reciprocal index; and population regularity or equitability—derived from the Simpson index) using MOTHUR software. Non-metric multidimensional scaling (nMDS) plot, homogeneity of molecular variance (HOMOVA) and permutational multivariate analysis of variance (PERMANOVA) using Bray–Curtis dissimilarity matrix and permutations of 10,000 were performed to evaluate beta diversity. The variance in the microbial profiles for the two groups was compared with an analysis of molecular variance (AMOVA) test (with 10,000 permutations) [35]. Beta dispersion was assessed with the homogeneity of molecular variance (HOMOVA) test for homogeneity of variance between the two groups (10,000 permutations) in MOTHUR [36,37]. Non-metric multidimensional scaling analysis (NMDS), based on the Bray–Curtis dissimilarity matrix, was applied to visualise biodiversity between groups [38]. Ordination analysis and 3D graphics were performed with Vegan (https://CRAN.R-project.org/package=vegan, accessed on 24 October 2021), Vegan3d (https://CRAN.R-project.org/package=vegan3d, accessed on 24 October 2021) and rgl packages (https://CRAN.R-project.org/package=rgl, accessed on 24 October 2021) in R (R: A Language and Environment for Statistical Computing, R Foundation for Statistical Computing, Vienna, Austria, 2015; https://www.R-project.org/, accessed on 24 October 2021).

Differential abundances of bacterial population between groups were assessed with DESeq2, using Deseq2 package in R [39].

## 3. Results

### 3.1. Culture Results of the Selected Samples for Microbiota Sequencing

In total, nine isolates belonging to seven bacterial species were identified by bacteriology in the six positive samples. The number of bacteria identified in the positive samples varied between one and three bacteria with a median of one bacterium. The details of bacteriology results of the randomly selected cross-sections are displayed in Table 1.

### 3.2. Description of the Surgical Site Microbiota during the Elective CS Realisation

The study of the surgical site microbiota during the realisation of elective CSs in BBCB cows highlighted 2542 distinct bacterial operational taxonomic units belonging to 567 genera derived from 28 phyla. A total of 121,083 reads were used for the 13 analysed samples.

The most abundant bacterial phyla were *Tenericutes* (45%), *Firmicutes* (26%), *Actinobacteria* (15%), *Proteobacteria* (10%) and *Bacteroidetes* (4%), and the others were found at very low rates.

The most abundant genera were *Mycoplasma* (44%), *Corynebacterium_1* (9%), *Bacillus* (8%), *Acinetobacter* (6%), *Ruminococcaceae_UCG-010* (2%) and *Ruminococcaceae_UCG-005* (2%), and the others represented 1% or less of the identified genera. The most often identified genera in each sample are displayed in the Figure 2.

The most abundant species were *Mycoplasma_wenyonii* (36%), *Bacillus_licheniformis* (8%), *Mycoplasma_AY837724.1.1457* (7%), *Corynebacterium_1_JF181315.1.1341/JF181360.1.1341/JF181303.1.1339* (2%), *Corynebacterium_1_efficiens* (2%) and *Corynebacterium_1_MF092158.1.1482* (2%). All the others represented 1% or less of the identified bacteria species.

### 3.3. Evaluation of the Microbiota Differences between the Samples Positive and Negative to the Bacteriology

Alpha diversity analysis revealed no significant differences for the following indices: Simpson reciprocal biodiversity index, Chao1 richness estimator for bacterial genera and the population regularity index (Figure 3). The alpha diversities of the samples that were positive and negative to the bacteriology were similar.

The NMDS analysis showed no difference between the compositions of the surgical site microbiota of the positive and negative cross-sections in the bacteriology (Figure 4). In fact, the analysis of molecular variance (AMOVA) showed no significant difference between the samples positive and the negative to the bacteriology (F score: 0.764; *p*-value: 0.562). Moreover, the analysis of the homogeneity of molecular variance (HOMOVA) also showed no significant difference between these groups (BV0.059; *p*-value 0.657).

### 3.4. Comparison of the Results of Culture and Amplicon Sequencing

In total, the genera of bacteria species identified by culture represented 13% (8462/64,646) of the reads identified by amplicons in negative samples to bacteriology, and 2.98% (1687/56,437) of the reads in the positive cultures.

Overall, five out of nine genera were identified by amplicon sequencing in the same sample in which bacteria species belonging to these genera were identified by culture. In fact, the genera *Psychrobacter* and *Pantoea* were never identified by amplicon sequencing in the same samples in which *Psychrobacter sp.* and *Pantoea agglomerans* were cultured. Moreover, the genera *Aerococcus* was not found by amplicon sequencing in two of three samples in which *Aerococcus viridans* was cultured.

We also observed that the bacterial species identified by culture were not frequently identified by amplicon sequencing. In fact, only *Acinetobacter sp.* and *Pseudomonas sp.* were identified by amplicon sequencing in the same sample in which they were cultured. The other species (seven of nine isolates) were not identified by amplicon sequencing in the samples positive to bacteriology. The comparison between the results of bacteriology and amplicon sequencing for the bacterial genera and bacterial species in the samples positive and negative to the bacteriology are displayed in Table 1.

In contrast, the most frequent genera (*Mycoplasma*, *Corynebacterium_1*, *Bacillus, Acinetobacter*, *Ruminococcaceae_UCG-010*, *Ruminococcaceae_UCG-005* and the others) and bacterial species found by amplicon sequencing (*Mycoplasma_wenyonii*, *Bacillus_liqueniformis*, *Mycoplasma_AY837724.1.1457* and others) were never found in the culture.

## 4. Discussion

To our knowledge, the current study is the first research investigating the microbiota composition of the surgical site during elective CSs in bovine medicine. Thus far, the microbiota assessment in the field of bovine obstetrics and reproduction aimed to describe the bacterial population of a gravid and virgin uterus, foetal fluids, digestive and respiratory systems of the foetus and the relation between the cow and calf microbiota and its impact on the new-born’s health [21,22,23,24,25,40]. The results of these studies unveiled unknown information about the microbiota of several tissues and organs previously considered sterile. Moreover, these studies were used to design, to criticise and to interpret the results of our research [21,23,24,41].

The lack of research dealing with the microbiota composition of surgical sites results from the difficulty to design an absolute study allowing a complete interpretation of the results. In addition, traditional hurdles faced within the study of the cattle’s microbiota are cows’ cost, housing and environmental sterileness [41]. The polyvalent causes of the surgical site contamination during the elective CS made it more challenging, especially to find the origin of the identified bacteria. The composition of the surgical site’s microbiota during the CS realisation might be influenced by the endogenous contamination, which could be provided by the foetal fluids [23,24], or by the haematogenous spread of cows’ bacteria [6,25]. In addition to that, the environmental bacteria could be an important source of contamination, since usually the CS is performed in unclean conditions [23,41,42]. The assessment of bacteria’s origin is more challenging, as the same bacteria might come from different sources of contamination [21,41]. Moreover, although it is strongly believed that the peritoneum is sterile [43], the microbiota composition of healthy peritoneum is never documented, indicating that the identified bacteria might be the normal flora of this serosa [43].

The surgical site is supposed to be sterile or with a very low amount of bacteria [28,29,44]. In contrast, bacteria were found in all samples examined by amplicon sequencing.

The most abundant genus is *Mycoplasma* (44%), mostly represented by *Mycoplasma wenyonii*. In fact, *Mycoplasma* species can invade several organs, tissues and mucosa. They can spread across the blood throughout the entire body. *Mycoplasma* species are also found in secretions (nasal discharge) and excretions (milk, colostrum and sperm) [45,46,47,48,49,50,51,52]. The massive presence of *Mycoplasma* genera in the surgical site of elective CSs might be related to the high prevalence of *Mycoplasma* species in Belgian livestock. Indeed, *Mycoplasma wenyonii* was identified by PCR in sampled blood of 59 out of 60 herds tested in the south of Belgium (Wallonia) [53], the same area from which our samples were provided. Moreover, *Mycoplasma bovis* was identified in around a third of Belgian herds [50,51]. We strongly assume that the presence of *Mycoplasma* species in the surgical site is mainly due to the haematogenous spread following the high symptomatic and asymptomatic carriage in Belgium rather than exogenous contamination. Moreover, the stress endured by the cows in the last days of pregnancy support the reactivation of the latent Mycoplasma and its blood circulation to the surgical site [7,49,53]. Following these findings, we strongly assume that haematogenous spread is the principal mode of contamination of the surgical site during the elective CS realisation.

The other genera, such as *Corynebacterium_1* (9%), *Bacillus* (8%) and *Acinetobacter* (6%), represent a considerable proportion of the microbiota encountered in the surgical site of elective CSs. In fact, they could be environmental, ubiquitous and/or commensal flora of the mucosa, skin or membranes [54]. They are frequently identified in sick and/or healthy cows [55,56]. Although haematogenous spread is possible but rarely reported [57], we strongly believe that the most likely mode of surgical site contamination by these bacteria might be exogenous, either by the operator and/or by inadequate disinfection of the surgical site and/or a high infectious pressure in the stables within the CS realisation [5,28,42]. The air suction occurring during the peritoneum incision [28,29,42] might emphasise the exogenous contamination by aspirating germs inside the peritoneal cavity and the surgical site. According to these findings, more importance should be given to the asepsis and the condition of the CS realisation to reduce the exogenous contamination.

Finally, the endogenous contamination of the surgical site should be taken into consideration during the elective CS realisation. This might be secondary to the foetal fluid bacteria. Indeed, several bacteria identified in the surgical site are also highlighted in the foetal fluids and meconium of new-born calves [22,23,24,25,41].

The implication of these bacteria in the occurrence of post-operative complications is not studied. These bacteria might be normal flora of the peritoneum and protective for the cows [22,23,24,25,41]. In this study, none of the involved cows showed post-operative complications. However, no conclusion could be drawn, since all these cows had received an intramuscular penicillin injection at the end of the CS.

A difference was observed between the culture and microbiota sequencing results. Samples that showed a negative culture could not be considered as sterile. In fact, using amplicon sequencing, 100% of the samples negative to the culture were shown to contain a high number of bacterial species DNA [58]. Moreover, the bacteria highlighted by culture are not the most frequently identified by amplicon sequencing, since the culture medium supported only the growth of some species [14]. In addition to that, the bacterial culture conditions do not allow the growth of *Mycoplasma* species [58], which were the most abundant bacterial population identified by the amplicon sequencing.

Surprisingly, seven out of nine of the bacteria isolates were not identified by amplicon sequencing in the samples in which they were cultured. This observation has rarely been reported in the studies comparing the amplicon sequencing and the culture results. In fact, the cultured bacteria are usually identified in the microbiota composition [14,58,59]. The assumptions which could explain our results are the possible misdiagnosis by the MALDI-TOF or sample contamination during the culture manipulations leading to false-positive culture [59,60]. Furthermore, it is likely that the bacterial load present in the sample may be below the analytical threshold of the amplicon sequencing test, which results in a false negative from the microbiota analysis [59]. In addition to that, the bacterial load and composition might be different between the swab sent to bacteriology and those sent to amplicon sequencing, even though they were taken at the same spot in the same moment. Nevertheless, further investigations are mandatory to answer this issue with accuracy.

Our study could be better designed if a negative control group was considered, especially to assess the absence of bacterial DNA of the used sampling material, even though it is supposed to be sterile [23,24]. However, several studies have not confirmed the sterility of the used material [40,56,61,62] without it interfering with the interpretation of their results. Moreover, if several samples had been taken at different sites before the uterus incision [22,23,24,25], such as the veterinary gloves, skin of cows, muscular wound and foetal fluids, this might have aided to the assessment of the origin of the bacteria found in the surgical site. However, the authors still consider this research a pioneer study describing the microbiota composition of the surgical site during the elective CS realisation. Therefore, it paves the way for further studies allowing to resolve the issues raised by this research.

## 5. Conclusions

Independent of the results of culture, all the sampled swabs showed the presence of a high number of bacterial species DNA (amplicon sequencing) in the surgical site of elective CSs. In contrast to the culture results, the majority of bacteria identified by amplicon sequencing belong to the *Mycoplasma* genus, most likely spread to the surgical site through haematogenous means. Surprisingly, a difference was observed between the results of the culture and those of amplicon sequencing, indicating that the cultured bacteria are not the most dominant species in the sample in which they were grown. This study raises the possibly for future research perspectives to investigate whether the identified bacteria are involved in post-operative complications or if they are normal flora of the peritoneum and surgical site of elective CSs.

## Figures and Tables

**Figure 1 vetsci-10-00094-f001:**
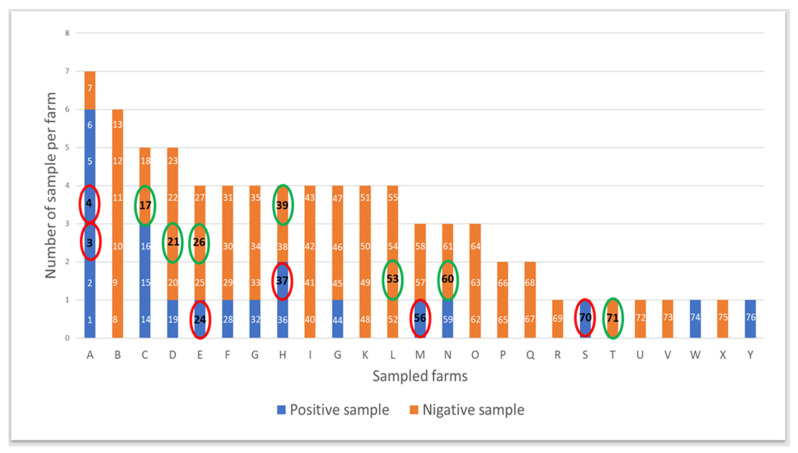
Samples selected randomly (13 samples: 6 among positive cultures circled in red and 7 among negative cultures circled in green) from the 76 swabs subjected to the bacteriology.

**Figure 2 vetsci-10-00094-f002:**
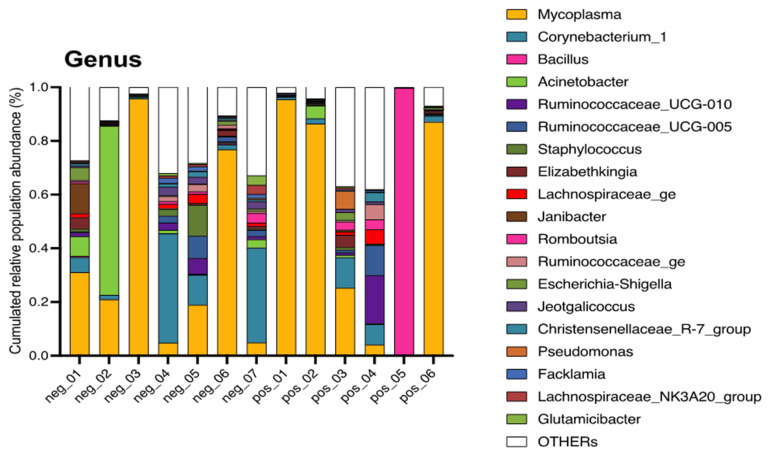
Genera with a mean relative abundance above 0.5% regarding the 13 cross-sections taken in surgical site during the elective caesarean section realisation in Belgian blue cows.

**Figure 3 vetsci-10-00094-f003:**
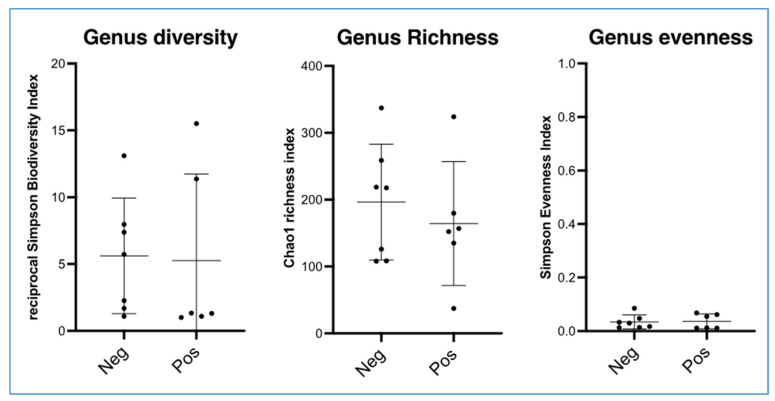
Alpha diversity, richness and regularity of bacterial genera in the surgical site during the caesarean section realisation in two groups of Belgian blue cows showing positive and negative bacterial culture. Data are scatter dot plots at the genus level for individual cows in the two defined groups with the mean and standard deviation.

**Figure 4 vetsci-10-00094-f004:**
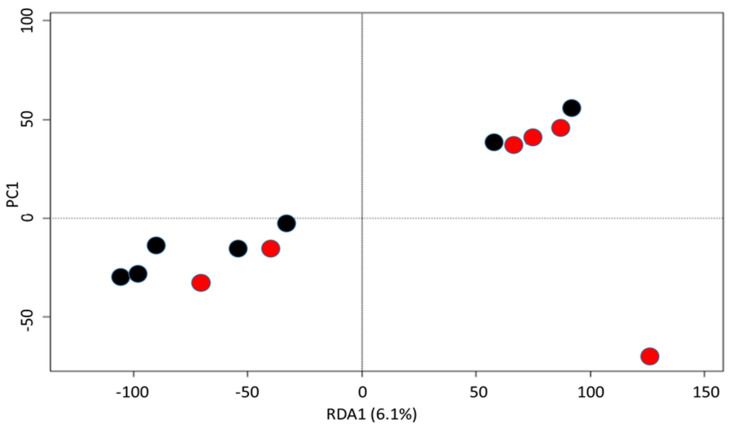
Beta diversity of the bacterial community microbiota identified in the surgical site during elective caesarean section in positive (red) and negative (black) culture samples. Multidimensional non-metric scaling (NMDS) plots generated by Bray–Curtis dissimilarity distances (model stress = 0.2122) and JACCARD matrix (model stress = 0.2774) were used.

**Table 1 vetsci-10-00094-t001:** Comparison between the result of culture and amplicon sequencing (genera and species) in the samples showing positive and negative bacterial culture.

Genera	*Aerococcus*		*Psychrobacter*		*Acinetobacter*		*Pantoea*		*Staphylococcus*		*Clostridium*		*Pseudomonas*	
Method of identification	Culture	Amp seq	Culture	Amp seq	Culture	Amp seq	Culture	Amp seq	Culture	Amp seq	Culture	Amp seq	Culture	Amp seq
Samples with culture (+)														
A3	+	-	-	+	-	+	-	-	-	+	-	+	-	+
A4	+	-	+	-	-	+	-	-	-	-	-	-	-	+
E24	-	+	-	+	+	+	-	-	-	+	-	+	-	+
H37	+	+	-	+	-	+	+	-	+	+	-	-	-	+
M56	-	+	-	-	-	+	-	-	-	+	+	+	-	+
S70	-	+	-	-	-	+	-	-	-	+	-	+	+	+
Samples with culture (−)														
C17	-	+	-	+	-	+	-	+	-	+	-	+	-	+
D21	-	+	-	+	-	+	-	-	-	+	-	+	-	+
E26	-	+	-	+	-	+	-	+	-	+	-	-	-	+
H39	-	+	-	+	-	+	-	-	-	+	-	+	-	+
L53	-	+	-	-	-	+	-	+	-	+	-	+	-	+
N60	-	+	-	+	-	+	-	+	-	+	-	+	-	+
T71	-	+	-	+	-	+	-	-	-	+	-	+	-	+
Total of positive samples	3/13	11/13	1/13	9/13	1/13	13/13	1/13	4/13	1/13	12/13	1/13	10/13	1/13	13/13
Species	*Aerococcus viridans*		*Psychrobacter* sp.		*Acinetobacter* sp.		*Pantoea agglomerans*		*Staphylococcus lentus*		*Clostridium perfringens*		*Pseudomonas* sp.	
Method of identification	Culture	Amp seq	Culture	Amp seq	Culture	Amp seq	Culture	Amp seq	Culture	Amp seq	Culture	Amp seq	Culture	Amp seq
Samples with culture (+)														
A3	+	-	-	+	-	+	-	-	-	-	-	-	-	+
A4	+	-	+	-	-	+	-	-	-	-	-	-	-	+
E24	-	-	-	+	+	+	-	-	-	-	-	-	-	+
H37	+	-	-	+	-	+	+	-	+	-	-	-	-	+
M56	-	-	-	-	-	+	-	-	-	-	+	-	-	+
S70	-	-	-	-	-	+	-	-	-	-	-	-	+	+
Samples with culture (−)														
C17	-	-	-	+	-	+	-	+	-	-	-	-	-	+
D21	-	-	-	+	-	+	-	-	-	-	-	-	-	+
E26	-	-	-	+	-	+	-	+	-	-	-	-	-	+
H39	-	-	-	+	-	+	-	-	-	-	-	-	-	+
L53	-	-	-	-	-	+	-	+	-	-	-	-	-	+
N60	-	-	-	+	-	+	-	+	-	-	-	-	-	+
T71	-	-	-	+	-	+	-	-	-	-	-	-	-	+
Total of positive samples	3/13	0/13	1/13	9/13	1/13	13/13	1/13	4/13	1/13	0/13	1/13	0/13	1/13	13/13

Culture: bacterial culture; amp seq: amplicon sequencing; blue: samples with positive culture and negative amplicon sequencing; yellow: samples with negative culture and negative amplicon sequencing; green: samples with positive culture and positive amplicon sequencing; not-coloured: Samples with negative culture and positive amplicon sequencing.

## Data Availability

All data are available in the manuscript.

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
