# Peer review of "Bacterial Contamination of the Surgical Site at the Time of Elective Caesarean Section in Belgian Blue Cows—Part 2: Identified by 16Sr DNA Amplicon Sequencing"

_vetsci, 2023, doi:10.3390/vetsci10020094_

Round 1
Reviewer 1 Report
The authors present a culture-independent study approach to survey bacterial contamination at a surgical site of Belgian blue cows. I think the information presented here does not stand alone to be a paper. This data should have been added to the culture-dependent paper which the authors indicate it is published already. That should have made the work complete. There are some corrections to be made through the manuscript (for example there are many places where the names of genera are not in italic).
Author Response
Please find attached the corrections.

Reviewer 2 Report
The manuscript is well written, the results presented in a clear way. The tests revealed a huge difference between the results obtained in both methods (cultura/genetic analysis) and showed that samples that were classified as microbiologically negative in the standard method turned out to contain DNA of many different microorganisms. This may be important in the context of preventing the development of a possible infection at the cesarean section site.
L30 - Mycoplasma - change into capital letter
L54 - Mycoplasma- italics
I think the names of figures and tables should always be capitalized, e.g. Figure 1 - L105, 131182, 191
L300 - MALDI-TOF
Author Response
Please find attached the corrections

Reviewer 3 Report
Dear Authors,
your manuscript reports an interesting study. However, some deficiencies are present and therefore some parts have to be revised. In details:
1)the aim of the study has to be better clarified and described;
2) bacteriological analyses should be carried out also on media specific for mycoplasma;
3) you considered the cultures as positive when one-four bacteria species were found and contaminated when more than four were cultured. This is an hypothesis, because for example also the isolation of three species could mean contamination; moreover , some species could not develope on the employed media. Consequently, results and discussion have to be revised.
4) In conclusion, you wrote "These findings indicate that the surgical site of elective CS is always contaminated" . The result is not so innovative.
Author Response
Please find attached the corrections

Round 2
Reviewer 1 Report
The authors addressed the issues presented.
Reviewer 3 Report
I think that your manuscript is worthy to be accepted in the current form.